# Patterns of satellite tagged hen harrier disappearances suggest widespread illegal killing on British grouse moors

Megan Murgatroyd [1], Stephen M. Redpath[1,2], Stephen G. Murphy[3], David J.T. Douglas[4], Richard Saunders [3] & Arjun Amar [1]

Identifying patterns of wildlife crime is a major conservation challenge. Here, we test whether deaths or disappearances of a protected species, the hen harrier, are associated with grouse moors, which are areas managed for the production of red grouse for recreational shooting. Using data from 58 satellite tracked hen harriers, we show high rates of unexpected tag failure and low first year survival compared to other harrier populations. The likelihood of harriers dying or disappearing increased as their use of grouse moors increased. Similarly, at the landscape scale, satellite fixes from the last week of life were distributed disproportionately on grouse moors in comparison to the overall use of such areas. This pattern was also apparent in protected areas in northern England. We conclude that hen harriers in Britain suffer elevated levels of mortality on grouse moors, which is most likely the result of illegal killing.

[1] FitzPatrick Institute of African Ornithology, DST-NRF Centre of Excellence, University of Cape Town, Rondebosch, Cape Town 7701, South Africa. [2] School of Biological Sciences, Zoology Building, University of Aberdeen, Tillydrone Avenue, Aberdeen AB24 2TZ, UK. [3] Natural England, Dragonfly House, 2 Gilders Way, Norwich NR3 1UB, UK. [4] RSPB Centre for Conservation Science, RSPB Scotland, 2 Lochside View, Edinburgh Park, Edinburgh EH12 9DH, UK. Correspondence and requests for materials should be addressed to M.M. (email: verreauxs@gmail.com) or to S.M.R. (email: s.redpath@abdn.ac.uk) or to A.A. (email: arjun.amar@uct.ac.za)

Wildlife crime represents a major threat to global biodiversity[1–4]. Many protected vertebrates, invertebrates and plants are targets for illegal activities when they have economic value[5–7] or when they are perceived to threaten livelihoods[8–10]. The effects of such activities can be profound[2,6].

One of the challenges in tackling this threat lies in identifying the true extent and patterns of these inherently secretive crimes[11,12]. Estimates for levels of wildlife crime have been derived in various ways, for example, using customs hauls to assess ivory poaching, DNA forensics, market surveys for wildlife trade or specialised questioning techniques, such as randomised responses with key stakeholders[13–16]. However, newer technological approaches, such as movement transmitters and remote sensing through drones and satellite imagery are likely to play an increasingly important role in identifying patterns of illegal wildlife crime in the future[17,18].

Modern tracking devices have the potential to support the development of strategies to reduce the extent of illegal activity, through, for example, identifying hot spots[9,11,19,20]. Devices that provide locational data have been used to track animals and identify instances of illegal killing[21,22]. Other research has explored spatial patterns in suspicious Global Positioning System (GPS) tag disappearances to highlight associations between suspected illegal activity and certain types of land use[20].

In this study, we explored the utility of satellite-tracking devices for understanding the extent and pattern of deaths and disappearances of the hen harrier Circus cyaneus. This raptor is protected under Annex 1 of the EU Birds Directive (2009/147/ EC)[23] and Schedule 1 of the Wildlife and Countryside Act 1981 (as amended)[24]. Hen harriers, along with other birds of prey, sit at the centre of a long-term, acrimonious conflict between conservationists and shooting interests in the UK[25–27]. Hen harriers are predators of red grouse Lagopus lagopus scotica and at the heart of this conflict lies the fact that predators such as harriers are illegally killed to provide larger post-breeding surpluses of red grouse that are recreationally shot[28]. The full extent and impact of illegal killing is unknown, yet it has been argued that persistent illegal activity is responsible for the current very low numbers of breeding hen harriers in England and their constrained population size and range in other parts of the UK[29,30]. Quantifying the extent and patterns of such behaviour will be key to the development of long-term sustainable solutions to this problem.

Here we bring together data from satellite-tracking devices and remotely sensed habitat and land management data to test the hypothesis that the patterns of deaths and disappearance of hen harriers are associated with land managed for grouse shooting (grouse moors). We find an association between the death or disappearance of tracked hen harriers and the use of grouse moors both at the individual level (i.e. harriers used grouse moors more than usual during the week preceding their death or disappearance) and at the landscape level (i.e. locational fixes during the week preceding death or disappearance are distributed disproportionately on grouse moors compared to other fixes). Hen harriers in this study also had a lower than expected first-year survival rate in comparison to other studies on similar species. We conclude that the increased likelihood of mortality is associated with illegal killing of this species on grouse moors.

## Results

### Hen harrier fates and first-year survival.
In total, 60 fledgling hen harriers (21 males, 39 females) were fitted with satellite transmitters between 2007 and 2017. Two transmitters were classified as having failed at the outset and were not included in the analyses. We therefore analysed data from 58 birds, using data collected up until 5 October 2017 (Supplementary Table 1). The fate of each tagged bird was assigned to one of the five categories (Table 1): A—alive, N—died of natural causes (confirmed), I— died of illegal activity (confirmed), TF—tag failure due to malfunction (confirmed or suspected based on transmitter data), and SNM—stopped no malfunction (where the transmitter stopped abruptly and unexpectedly based on diagnostic plots and the bird was never found)[20]. Seven birds were still alive at the data cut-off point (A—12%). Five birds were recovered and autopsies confirmed they died of natural causes (N—9%). Three birds were recovered dead and autopsies confirmed that they were illegally killed and one additional tag was recovered with the harness intact but without the bird's body indicating an illegally killed bird (I—7%). Two birds were re-sighted after their tags had failed due to a malfunction (the individuals were recognisable by photography of the metal identity ring or patagial tag), and an additional two were classified as having failed due to a malfunction following an examination of their diagnostic plots (TF— 7%). Three of the four tag failures occurred after the first year of life. All other birds were classified as having tags that stopped transmitting with no indication of a malfunction ($n = 38$; SNM— 66%). Thus 42 birds (72%) were either confirmed to have been illegally killed or disappeared suddenly with no evidence of a tag malfunction. Three harriers, tagged in 2017, were only tracked for 12–14 weeks (and were still alive at the data cut-off point), and one tag failure occurred within 2 weeks; these individuals do not have sufficient data to estimate first-year survival. In total, 45 harriers (of 54 with sufficient first-year tracking period) died or disappeared during the first year (365 days since tagging), giving a first-year survival rate of 17%. Most of the deaths during the first year occurred during the late summer and autumn period, within 20 weeks of fledging ($n = 34$, 76%, Supplementary Fig. 1). Only 20 individuals lasted through the first 20 weeks. During this period, the mean percentage of fixes on grouse moors per week for harriers that survived was (±SE) 15 ± 2.6 %, which was half of the mean percentage for those which died or disappeared (30 ± 3.9 %).

### Use of grouse moors during the terminal week of life.
The number of fixes per bird per week averaged (±SE) 16.6 ± 0.3. Harriers were more likely to be located on grouse moors during the terminal week (i.e. the last 7 days of tracking prior to the date of death or disappearance) than during other weeks (Fig. 1a). Moreover, the probability of a bird dying or disappearing increased with the proportion of fixes on grouse moors (Type II Wald chi-square: $X^2_{1, 1471} = 4.066$, $p = 0.044$) and this pattern was more pronounced when only data from tracked birds that were known to have been illegally killed and those with tags that were classed as SNM were tested (Fig. 1b. $X^2_{1, 272} = 8.832$, $p = 0.003$). During the terminal week of tracking, the home range size (median 95% Minimum Convex Polygon (MCP) ± SE) was 16 ± 47 km².

### Distribution of terminal fixes across the landscape.
At the landscape scale, the proportion of fixes that were from the terminal week was significantly associated with the percentage of each grid square with grouse moor (Fig. 2. All data, Type II Wald chi-square: $X^2_{1, 307} = 45.749$, $p < 0.001$). The same pattern was present when only data from tags classed as I and SNM were used (Fig. 2. $X^2_{1, 242} = 35.572$, $p < 0.001$). Fixes from the terminal week were distributed disproportionately on grouse moors compared to their overall use (Supplementary Fig. 2). The proportion of fixes in each 20 × 20 km² grid square, attributed to terminal weeks varied from 0.02 in grid squares with no grouse moors to 0.20 in squares with 50% grouse moor, indicating that harriers were ten times more likely to die (I and N) or disappear (SNM) in areas

| Table 1 Summary of fate classifications from 58 satellite-tracked hen harriers | | |
|---|---|---|
| **Classification** | **Description** | **No. of birds ascribed to each category (%)** |
| A | Bird alive and tag still transmitting | 7 (12%) |
| N | Bird recovered and cause of death established to be natural | 5 (9%) |
| I | (i) Bird dead, confirmed to have been illegally killed or (ii) tag harness recovered intact with no evidence of bird | 4 (7%) |
| TF | (i) Transmitter malfunctioned (i.e. the tag ceased transmitting but the bird was seen alive) or (ii) tag failure likely due to diagnostic plots | 4 (7%) |
| SNM | Transmitters suddenly stopped with no malfunction detected | 38 (66%) |

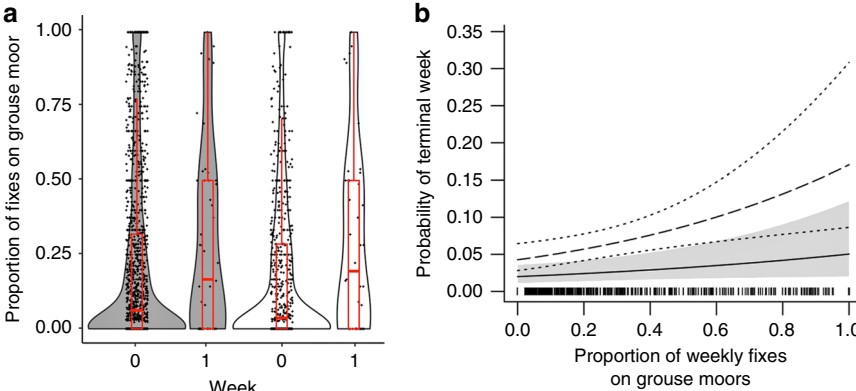

**Fig. 1** Use of grouse moors in relation to terminal weeks. **a** The density distribution and data spread for the weekly proportion of fixes on grouse moors during live weeks (0) and terminal weeks (1) for all tracked hen harriers (grey shaded, $n = 58$ individuals) and for harriers known to have been illegally killed (I) combined with those which had tags that suddenly stopped with no indication of a malfunction (SNM) (white, $n = 42$ individuals). Box plots (red) show the median, upper and lower quartile and whiskers (1.5× interquartile range). **b** The effect of grouse moor use on the probability of it being a terminal week (i.e. the probability of death or disappearance). Solid line represents all harriers with confidence intervals (CIs) shaded grey; dashed line represent I and SNM harriers only, with CIs shown by dotted lines. Source data are provided as a Source Data file

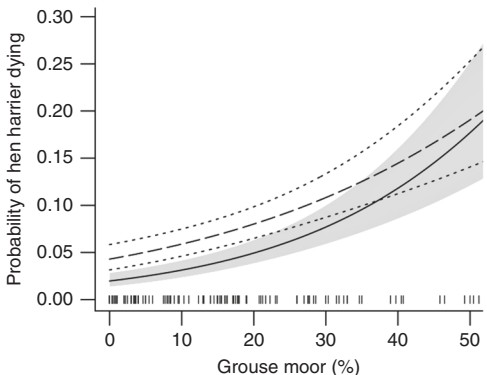

**Fig. 2** Probability of hen harrier dying in relation to grouse moors. The percentage of grouse moor habitat is calculated from a 20 × 20 km$^2$ grid in the UK mainland and Isle of Man. The probability of a hen harrier dying is derived from a generalized linear model testing the proportion of fixes from hen harriers' terminal weeks. Solid line represents all harriers with confidence intervals (CIs) shaded grey; dashed line represent data from harriers that were known to have been illegally killed (I) and those with tags that stopped suddenly with no prior indication of a malfunction (SNM) only, with CIs shown by dotted lines. Source data are provided as a Source Data file

dominated by grouse moors (Fig. 2). Using only fixes obtained from tags classed as I and SNM in northern England and southern Scotland (an area that encompassed 91% of the tracking data and 98% of all I and SNM data), we found that squares where harriers had a higher than average likelihood of dying or disappearing were associated with the highest percentage of grouse moor coverage (Fig. 3). This pattern was evident when comparing the proportion of fixes from live weeks and terminal weeks in relation to the cover of grouse moor in squares (Fig. 3b). Of the live fixes, 0.14 were located in squares with >23% grouse moors, compared to 0.38 of the fixes from terminal weeks. In summary, harriers had a much higher likelihood of dying or disappearing in squares with the most grouse moor coverage. A small number ($n = 6$) of grid squares outside of areas managed for grouse moors also had a high proportion of terminal fixes. In most cases, these squares directly bordered squares with managed grouse moors ($n = 5$), and it is likely harriers were moving between squares. Only one square had a high proportion of terminal fixes but did not directly border a grouse moor and this can be attributed to the movements of just one individual.

**The proportion of terminal fixes in English protected areas (PAs).** As the percentage area of grouse moor within a PA increased, there was an increase in the proportion of terminal fixes per PA (Supplementary Fig. 3, Type II Wald chi-square: $X^2_{1,6} = 9.837$, $p = 0.002$). This pattern was unchanged when tested only on harriers known to have been illegally killed (I) and those that disappeared suddenly (SNM) (Supplementary Fig. 3, $X^2_{1,6} = 9.944$, $p = 0.002$). This suggested that harriers were more likely to be illegally killed in PAs that had more grouse moor habitat. For those birds that were illegally killed or disappeared, the North York Moors and the Peak District followed by the North Pennines, Nidderdale, Yorkshire Dales and Forest of Bowland had the highest proportion of terminal fixes, indicating higher than expected harrier mortality in relation to use (Fig. 4).

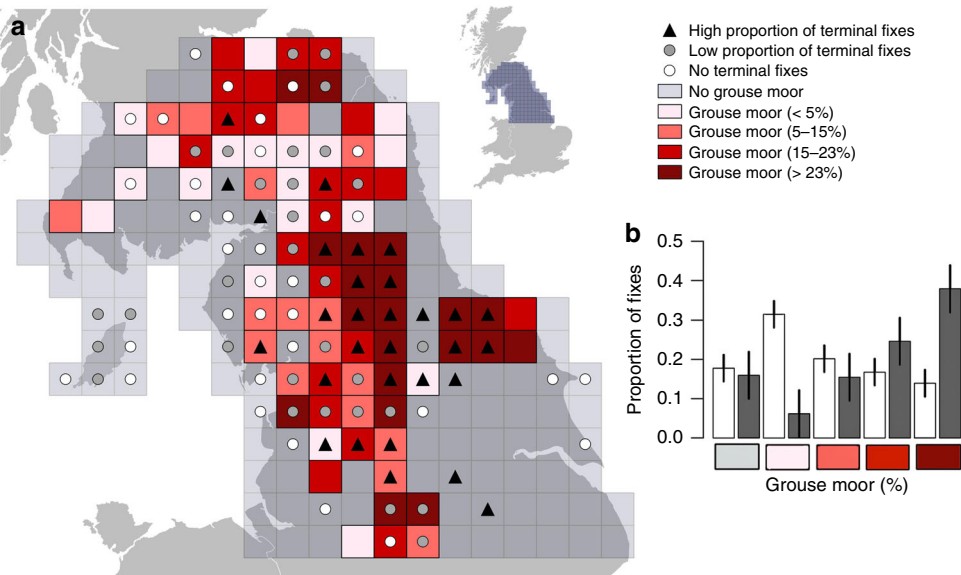

**Fig. 3** Hen harrier terminal week fixes in relation to grouse moors. Data include only satellite-tracked hen harriers that were known to have been illegally killed and those that suddenly disappeared when their tag stopped with no indication of a prior malfunction ($n = 42$) and the area (grid squares) encompassed 98% of the tracking data from these individuals and 91% of data from all tracked harriers. **a** Data in northern England and southern Scotland are displayed on a $20 \times 20$ km$^2$ grid. Points (circles and triangles) are displayed for all grid squares with more than five fixes. White circles show grid squares used by hen harriers with no fixes from terminal weeks. Grey circles represent grid squares with a below average (median) proportion of terminal week fixes and black triangles represent above average proportion. Grouse moor distribution is shown (red scale) and calculated as the percentage of 1-km grid squares per 20 km square with heather burning (grouse moor management, Douglas et al.[59]). **b** The graph shows the proportion of fixes (±SE) that fall into each grouse moor group (%) from live weeks (light bars) and terminal weeks (dark bars)

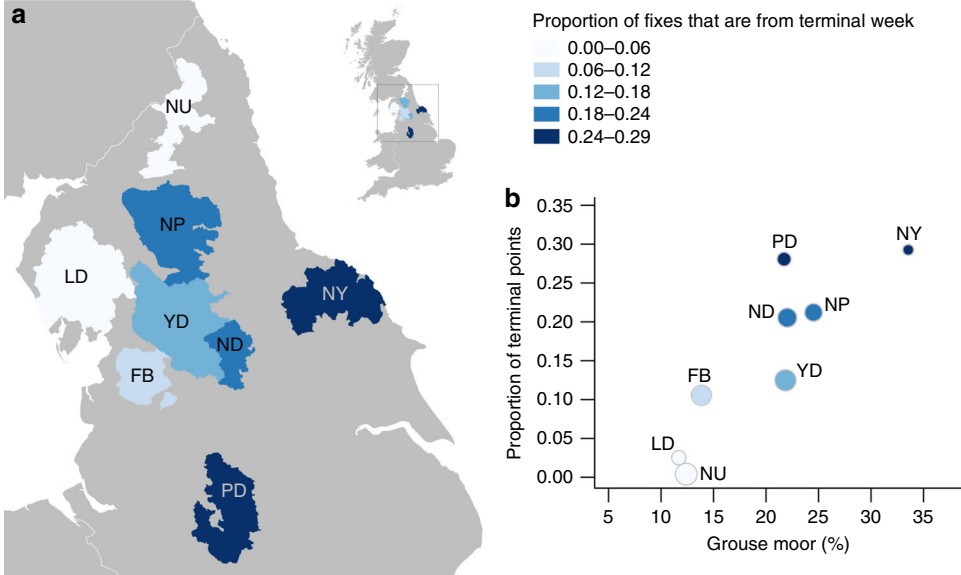

**Fig. 4** Hen harrier terminal week fixes in relation to protected areas (PAs). **a** Map showing the proportion of all fixes that were in the terminal week of tracking from hen harriers fitted with satellite tags, in relation to National Parks and Areas of Outstanding Natural Beauty in the north of England PAs. Data includes only hen harriers that were known to have been illegally killed and those that suddenly disappeared when their tag stopped with no indication of a prior malfunction ($n = 42$). **b** The association between the proportion of terminal fixes per PA and the percentage of PA managed for grouse moors. Symbol size represents the relative sample size of fixes per PA. NY North York Moors, PD Peak District, NP North Pennines, ND Nidderdale, YD Yorkshire Dales, FB Forest of Bowland. LD Lake District, NU Northumberland. Contains public sector information licensed under the Open Government Licence v3.0

## Discussion

Of the 58 satellite-tracked hen harriers, 4 were found with direct evidence of illegal killing, while 38 simply disappeared; the tags suddenly stopped transmitting without any prior evidence of tag malfunction, no remains could be found and the birds were not seen again. Thus, despite the lack of physical evidence, this strongly suggests destruction of the tag and removal of the carcass. We conclude that illegal killing is the most parsimonious explanation for the fate of these birds.

Three lines of evidence provide strong support for the hypothesis that illegal killing was associated with land managed for grouse shooting. First, the low first-year harrier survival rate of

17% is likely to be explained by illegal killing. This was lower than equivalent estimates derived from re-sightings of wing-tagged harriers elsewhere. In the Scottish Orkney Islands, where there is no managed grouse moor, but where some birds do disperse to mainland Britain, survival was c. 37% for first-year males and c. 54% for females[31,32]. In the Scottish mainland population, where illegal killing also occurred, survival was 36%[28]. In Spain, the survival rate of first-year Montagu's harriers *Circus pygargus* was estimated to be c. 50%[33]. Second, individual harriers spent more time on grouse moors in their terminal week than in previous weeks. Individuals tagged as nestlings in northern England and southern Scotland dispersed around the country after fledging, but the likelihood of them disappearing increased significantly when they spent more time on grouse moors. Third, the likelihood of harriers dying or disappearing increased in landscapes and even PAs where more of the land was managed for grouse.

We can think of no alternative, plausible explanation as to why mortality and unexpected tag failure was occurring at a higher rate on grouse moors. There are no other obvious potential sources that would be expected to yield such high mortality on managed grouse moors. Predators such as golden eagles *Aquila chrysaetos* and red foxes *Vulpes vulpes* are scarce and we would expect to recover tags from naturally predated harriers[34,35]. Hen harrier collisions with wind turbines are relatively rare and again we would expect remains of hen harriers to be recovered in post-construction monitoring of such site, so collision mortality seems unlikely to explain the results[36]. Harrier prey abundance and prey capture rates for hen harriers have been found to be higher on grouse moors than other habitat types[28,37]. This suggests that starvation is an unlikely source of mortality on grouse moors. In addition, even if mortality was caused by something other than illegal killing, we would expect to find at least some of the tags from the birds that disappeared as was the case for natural mortalities. Our inference that illegal killing is responsible for these losses matches previous studies that have highlighted illegal killing of hen harriers and other raptors on grouse moors[8,28,38–41].

Tag failure rates can be difficult to detect, particularly for Doppler tags like the ones used in this study, and this can add noise to analyses of mortality dynamics[42]. Where tag failures were identified in this study, it was clear from the diagnostic plots that these tags had been working abnormally prior to cessation of tracking[20,43]. Where tags stopped abruptly and unexpectedly, field staff searched the areas of the last known fixes extensively and sometimes repeatedly. Carcasses were rarely recovered, presumably due to suspected illegal killing and carcass disposal, and there were no observations of live hen harriers from the SNM group with failed tags, as would be expected in some cases if the tags from these birds had actually failed and the birds were still alive. Furthermore, our recorded tag failure rate (7%) is very similar to that found in a study on Montagu's harriers (6%), using the same type of Doppler tags manufactured by the same company[43] (Klaassen, R., pers. comm.). Thus, although it is impossible to ascertain if all of our tags classified as SNM were cases of illegal killing, any erroneous non-persecution events included would only serve to add noise to our analyses and thus any estimates would be conservative. Despite this potential uncertainty, we still detected patterns in all analyses for significant associations between death or disappearance of hen harriers and use of grouse moors.

One of the challenges of detecting wildlife crime, even with the use of satellite telemetry, is that illegal behaviour is naturally hard to detect because those involved are likely to destroy incriminating evidence[20,44,45]. Indeed, the only incidences where illegal activity was confirmed via recovery of carcasses and their subsequent post mortem was for three birds that were shot in the legs

and were likely to have been still capable of flight and with functioning transmitters[46,47]. Our ability to find tags was limited by the type of tags used here. The duty cycle meant that we were unable to detect the exact time and location of death if tags stopped working. New tags with more frequent upload cycles and more regular fixes may help pinpoint illegal acts much closer to the time when they actually occur, especially given the likelihood of tag destruction. Real-time, anti-poaching transmitters, as proposed for deployment to reduce the illegal killing of large mammals[48] might also improve detection probabilities. Also, additional tag functions such as geofencing, whereby real-time locations can be sent when tracked animals enter a pre-defined area of known higher risk[49] will be useful if they become sufficiently miniaturised for tracking harriers. At the same time, increased surveillance from new forms of technology to monitor wildlife crime brings a need to ensure that data collected are used responsibly, to avoid potential negative social impacts and infringements of privacy[50,51]. Any such ethical considerations arising from the use of surveillance to monitor illegal behaviour need to be integrated into the application and governance of these new technologies.

Our analyses identified PAs in England where suspected illegal killing was prevalent. Unlike in many other parts of the world, national parks in the UK are heavily modified by human activities rather than representing wilderness areas and are more often designated for their aesthetic value rather than the contribution they are expected to make to biodiversity conservation. Here PAs that contained more managed grouse moors were more likely to be the places where suspected illegal activity was occurring. Thus, within the PA network, we have identified those areas where priorities for actions to reduce illegal killing of raptors might be most usefully directed.

This study strengthens the evidence base on the extent of the illegal activity on grouse moors by empirically evaluating the fate of a large sample of tracked birds and can thus inform action to deal with this problem. However, there is currently much debate over what those actions should be[52]. Some call for driven grouse shooting to be banned[30], or licenced[27,53], or for the promotion of alternative grouse shooting cultures that do not rely on intensive management[27,53]. At the same time, the UK government department DEFRA (Department for Environment, Food and Rural Affairs) and their agency responsible for conservation (Natural England) are working with the grouse shooting industry and some conservation NGOs to implement a management plan involving a variety of different activities, including protecting nests/roosts, re-introducing harriers to southern England away from grouse moorland and trialling the temporary licensed removal of young birds from grouse moors[54]. Views vary widely as to which of these approaches will provide the best long-term solution and help overcome this persistent and damaging conflict. This discussion reflects wider debates over the focus on enforcement versus prevention to tackle environmental crime[11,12].

In conclusion, our analyses show that satellite-tracking data combined with remotely sensed data can be used to explore patterns of illegal killing and to pinpoint a land management type and areas associated with these illegal activities. This approach will likely prove useful in tackling a range of issues associated with the illegal killing of wildlife. As the technology of tracking devices advances, the likelihood of pinpointing such illegal activity is likely to increase, supporting wildlife crime detection and deterrence in the future.

## Methods

**Satellite transmitters**. To obtain birds for tagging, nests were located by trained staff and volunteers, who monitor traditional harrier breeding grounds and use observations of adult behaviour to pinpoint nest locations. A few nests ($n = 2$) were

also located from the location of fixes from previously tagged breeding birds. No nest selection process was taken to determine which chicks would be fitted with transmitters due to the small number of nests in England or the Isle of Man ($n =$ 31) or southern Scotland ($n = 14$), so tagging was opportunistic and either one or two chicks per brood were tagged. Nestlings were tagged mainly in northern England ($n = 36$), southern Scotland ($n = 19$) and the Isle of Man ($n = 5$) (exact locations are not given due to the sensitive nature of these data). Hen harrier nestlings were fitted with satellite transmitters (Microwave telemetry Inc, Columbia, MD, USA) between 2007 and 2017. Tags were fitted just prior to fledging, when nestlings were 26–32 days old[55]. All nestlings were fitted with individual identity BTO (British Trust for Ornithology) metal rings and some tracked individuals were also fitted with patagial tags ($n = 5$). Transmitters weighed 9.5 g ($n = 52$), except for 12 g ($n = 7$) tags that were used on females (typically heavier than males) from 2009 to 2011 (and a single re-deployment in 2014). Transmitters were fitted using a backpack harness made from 6-mm wide Teflon ribbon (Bally Ribbon Mills, Bally, PA USA). As in similar tracking studies[43], we found no tags to have fallen off naturally.

The transmitters used in this study did not provide real-time locational data. Instead they were programmed to be on for 8 h and then off for 48 h ($n = 56$) or on for 10 h and then off for 60 h ($n = 4$). During a typical on period, up to 18 locations were sent via a remote upload to the Argos satellite system. Although the number of transmissions per day depended on season and weather conditions, this duty cycle allowed the transmitters' batteries to remain sufficiently charged (with at least two tags still functioning after 4 years) using their integrated solar panel. However, because locational data were only received every c. 3 days, we could not simply use the last known fix to infer the location where the bird died or disappeared. We therefore allocated the locational data into weeks, with the week prior to death or disappearance termed the terminal week.

These small transmitters did not have a built-in GPS but instead communicated with satellites as part of the Argos system. They used Doppler shift data to estimate the location and the accuracy of this location, which depended mostly on the number and distribution of transmissions received during a satellite pass. Each location was assigned an accuracy class (LC) of 1–3 or 0, A, B, Z, depending on the number of messages received per satellite pass, with the following levels of spatial/ location accuracy 3: <250 m, 2: 250–500 m, 1: 500–1500 m, 0: >1500. Locations with the class A, B and Z were acquired with ≤3 messages and thus lack accuracy estimations. Each fix was also associated with several other pieces of sensor information, including internal temperature of the transmitter and battery voltage. Transmitted data were archived at the Argos processing and archive centre (CLS, Toulouse, France), from which we obtained all data, except for data from four individuals which were obtained by the Hawk and Owl Trust during regular downloads and supplied to us directly.

**Ethical compliance**. The hen harrier is listed on Schedule 1 of the Wildlife and Countryside Act 1981 (as amended) and all potentially disturbing work affecting breeding birds, and/or their dependent young, was carried out under licence by experienced individuals. The use of harness-mounted radio and satellite tags was approved by the Special Methods Technical Panel of the British Trust for Ornithology's Ringing Committee. Tracking work, using harness-mounted transmitters, was also approved by Natural England's Executive and Non-executive Board.

**Classifying the fates of tagged birds**. Detecting instances of illegal killing is difficult because of the likelihood that those involved in such activity will destroy the tags on birds they kill before the location can be transmitted[20]. There is an expectation therefore that illegally killed birds would be hard to recover. Despite this, we were able to assign fates to all dead/disappeared birds on the basis of three types of information: (1) autopsies from all recovered birds, which allowed us to separate natural deaths from confirmed illegally killed birds; (2) re-sightings of birds with failed transmitters, which were recognisable from patagial tags, and finally (3) diagnostic information from tags that ceased tracking but were not recovered (to separate potential tag failures from tags that stopped transmitting unexpectedly with no indication of any malfunction)[43]. Transmitters can fail due to technical malfunctions, which may be preceded by a dropping of battery voltage or more intermittent or inaccurate fix transmissions or erratic temperature readings[20,43]. To explore whether a transmitter may have failed for technical reasons, we created four plots for each transmitter using all available fixes irrespective of accuracy class. We identified transmitters that were outside of the normal operating parameters (Supplementary Fig. 4) to identify those transmitters that may have failed owing to a malfunction. To do this, we examined battery voltage, fix interval and temperature for the time of year and scrutinized the mean daily distance travelled keeping in mind that larger distances travelled can be an artefact of poor quality locations. The fate of each tagged bird was then assigned to one of the five categories: Alive (A); died of natural causes (N); confirmed illegally killed (I); tag failed (TF); stopped no malfunction (SNM) (Table 1).

**Data processing**. A basic spatial filter was applied to remove outliers caused by inaccurate locational fixes. Longitude was constrained to between −10.2 and 2, latitude between 47 and 59.5. We removed any duplicate fixes, retaining only those fixes with the best quality LC (3, 2, 1, 0) and highest battery voltage. Where a tag's

last fixes were stationary (i.e. indicating the bird or tag was not moving for more than a day), multiple fixes were removed to ensure that they did not bias results. This generally only occurred for the few tags that were recovered (e.g. classified as I or N).

Weeks were assigned in reverse order; thus the 7 days prior to the date of the last fix (i.e. the terminal week) were assigned week 1. All other weeks were assigned 0. This ensured that data were maximised for the week prior to death or disappearance, which were the most relevant data for our analysis. For birds that remained alive (A) or for the few birds with confirmed tag failure (TF) (Table 1), all weeks were scored 0. We calculated the home range size as the 95% MCP for all terminal weeks with ≥5 fixes using the MCP function in adehabitatHR[56]. All analyses were performed in R version 3.3.0[57].

**Grouse moor data**. Grouse moors dominate large areas of the British uplands; they are largely treeless habitats dominated by the dwarf shrub heather *Culluna vulgaris*, which is the main food of adult red grouse[34]. Burning of heather in small strips (up to 2 ha) in rotations of 10–25 years is a ubiquitous moorland management practice to support red grouse shooting, detected over at least 8551 km$^2$ of mainland UK[34,58,59]. Post-burning regrowth is distinguishable from the surrounding unburned vegetation and the resultant patterns of burnt areas are easily detectable from remotely sensed images for up to ca. 25 years[8,59,60]. Such imagery has been used to describe the spatial pattern of land managed for grouse shooting across mainland UK at a 1 km scale[59], and these data were used in this study. We defined grouse moor as any 1 × 1 km$^2$ grid square with evidence of strip burning. For each harrier location, we determined whether or not it was in a grouse moor square. Fixes outside of the strip-burned areas[59] were regarded as not on grouse moor, which is readily apparent from aerial photographs[59,60].

**Statistical analysis**. For all analyses, we first included data from all tags (i.e. confirmed natural and illegal deaths, stopped no malfunction, live birds and failed tags, the latter two categories had no terminal weeks). We then repeated all analyses using only data from known illegal mortality (I) and from tags that unexpectedly stopped transmitting (SNM). All analysis are summarised in Table 2.

**Use of grouse moors during the terminal week of life**. The first analysis allowed us to compare the proportion of fixes that occurred on grouse moors in the terminal week to the proportion in all other weeks. This also indicated whether the probability of a harrier dying or disappearing on grouse moors was higher than expected from overall habitat use. We used a generalized linear mixed model (GLMM) with a logit-link function and a binomial error structure using the package lme4 in R[61]. The response variable was a binary code for each week where 1=terminal week of transmission and 0=all other alive weeks. The proportion of fixes on grouse moors for each week and the maximum number of days since tagging for each week were entered as fixed effects. Days since tagging was entered as the maximum value per week and included to control for a potential change in survival rate with age. Individual bird identity (birdID) was included in the model as a random term, to account for the non-independence of repeat fixes from individual birds. The explanatory variables were centred (by subtracting sample mean) and standardized (by dividing by standard deviation) to aid model convergence[62].

**The proportion of terminal fixes in the landscape**. To understand the spatial patterns, we tested whether harriers were more likely to die or disappear in landscapes with more grouse moors than expected from the relative use of those areas. This analysis was only conducted for the UK mainland and the Isle of Man, which contained >95% of the fixes. We first calculated the total number of fixes, irrespective of which bird they came from, for each 20 × 20 km$^2$ National Grid Square (amalgams from the 10 × 10 km$^2$ grid). We did this separately for terminal weeks and all other weeks combined. We then calculated the percentage of 1-km squares in each 20 × 20 km$^2$ Grid Square that contained grouse moor habitat. We used a binomial GLM using the cbind function to construct a two-vector response variable (number of terminal week fixes, number of non-terminal week fixes) to test if the response variable, which was equivalent to the proportion of the total number of fixes that were in the terminal weeks, was related to the fixed effect percentage of grouse moor habitat per square. This analysis is a weighted regression, thus the sample size per grid square is accounted for by using the total number of fixes in each square. Grid squares that contained no fixes at all were excluded from the analysis.

We then sought to identify the specific PAs where suspected illegal killing was most prevalent. We initially explored the idea of using special protection areas (SPAs), which are strictly protected sites classified in accordance with Article 4 of the EC Birds Directive (Directive 2009/147/EC)[23]. However, there are only two such sites classified for breeding hen harriers in the north of England, and sites classified for non-breeding hen harriers are restricted to lowland/southern England ($n = 16$). Their limited geographic coverage, in comparison to the extensive dispersal of satellite-tagged hen harriers across the English uplands, excluded the use of SPAs as the basis for this additional landscape-scale analysis. Thus, for this analysis, we used PAs including all National Parks and Areas of Outstanding Natural Beauty in northern England. We explored whether the proportion of the

**Table 2 Analyses summary**

| | Tracking data used | Tests | Response variable | Fixed effect | Random effect | Sample size |
|---|---|---|---|---|---|---|
| Association of last week with grouse moors | $n = 24{,}447$ $N = 10{,}974$, fixes | Binomial GLMM | Binary: terminal week (1) or an alive week (0) | —Weekly proportion of fixes on grouse moors —Days since tagging (max. per week) | Bird ID | $n = 1475$, $N = 692$ weeks |
| Landscape-scale analysis | $n = 23{,}357$, $N = 10{,}800$, fixes on UK mainland and Isle of Man | Binomial GLM | Two vector cbind: number of fixes in terminal week, number of other fixes, per 20 $km^2$ grid cell | Proportion of 1 $km^2$ cells with grouse moor cover in 20 $km^2$ grid | na | $n = 309$, $N = 244$ grid squares |
| Protected area analysis | $n = 5960$ $N = 4940$, fixes in protected areas in England | Binomial GLM | Two vector cbind: number of fixes in terminal week, number of other fixes, per PA | Proportion of 1 $km^2$ cells with grouse moor cover inside PAs | na | $n/N = 8$ protected areas |

Three main analyses used to explore whether death or disappearance of satellite-tracked hen harriers were associated with grouse moor habitat. Sample size of fixes used per analysis are described, where (n) includes fixes from all harriers and (N) is the sample size for repeat analyses using only suspected (SIM) and known (I) illegal mortalities

total number of fixes per PA that were from terminal weeks was associated with the percentage of 1 km squares within the PA that contained grouse moor. To do this, we filtered the tracking data to only include fixes within the boundary of each PA and fitted the number of fixes in the terminal weeks (1) and the number of fixes in all other weeks (0) in the same model as the previous analysis. PAs rarely used by the tracked hen harriers (<5 fixes) were removed.

**Reporting summary**. Further information on experimental design is available in the Nature Research Reporting Summary linked to this article.

## Data availability
The complete data sets analysed in this study are not publicly available due to the sensitivity of the locational data but are available from the corresponding author on reasonable request and with permission of Natural England. The source data for Figs. 1 and 2 have been provided as a Source Data file.

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

## Acknowledgements

Natural England commenced a Hen Harrier Recovery Project in 2002. This tracking study was funded exclusively by Natural England and is part of their on-going work on hen harrier conservation. We thank Hamish Smith and staff at the Hawk and Owl Trust for contributing data from four hen harriers they have tracked. We are grateful for the time of many volunteers in the field who monitored and searched for harriers: Pat Martin, Gavin Craggs, Pete Davies, Derek Hayward, Martin Davison, Mick Carroll, Paul Howarth, Ian Thomson, and Elsie Ashworth. We thank Judith Smith and Phil Skinner for sponsoring tags. Also we would like to thank the Wildlife Crime Officers in Lancashire, Yorkshire, Co Durham and Northumberland for their assistance. Thanks also to Jeremy Wilson and Pat Thompson for useful comments on this manuscript. We are grateful to staff at Microwave Telemetry Inc. and CLS France for data archiving.

## Author contributions

M.M., A.A. and S.M.R. conceived the study, analysed the data and wrote the manuscript. S.G.M. collected the data. D.J.T.D., R.S. and S.G.M. contributed to drafts of the manuscript and all authors gave final approval for publication.

## Additional information

**Competing interests:** The authors declare no competing interests.

