## [Peer Review File · Nature Communications]

Reviewers' comments:

Reviewer #1 (Remarks to the Author):

The paper examines the unaccountable disappearance of satellite-tracked harriers, and finds that most of the disappearances occur on grouse moors, where killing of harriers has long been suspected, accounting for the rarity of the species. This is an important paper, of interest to many people concerned with bird conservation and with the illegal killing of protected species. In my view the data are well analysed, with appropriate statistical treatment, and the conclusions are well justified.

My comments are mostly minor:

Line 145 reports re-sightings of birds with failed transmitters. Does this imply that birds were marked in other ways too? Explain.

Line 195. Can you say, how long after a burn, strip fires are apparent from an aerial view? Readers might think they can be discerned for only a year or two after a burn, when in reality I guess it could be more than a decade. This affects the validity of your method of defining locations of grouse moors.

Line 292. Can you say if any birds lasted a long time (say throughout the first winter) in a grouse moor area?

Line 344. Do you want to comment on the fact that some squares without grouse moors also showed relatively high terminal records? Is there anything you can say about these squares (eg wind turbines)?

Line 392. 'dispersed around the country' seems a fairly loose statement. Looking at the map, there seems to be a southerly bias, as no birds appeared above the Edinburgh-Glasgow line.

For other more minor comments, and suggestions on wording, see the copy of the ms attached.

Reviewer #2 (Remarks to the Author):

This is not a techniques paper but a paper identifying illegal killing of hen harriers. Certainly the study design does not allow a comparative analysis of the Argos collars or a systematic evaluation of radio performance. And such a techniques paper would warrant publication in Nature Comms, but this is simply a matter of the authors having lost their way and is easily fixed. Regardless, the results are important and we need such an analysis documenting the suspected killing of harriers by game keepers.

Line 18: Here we assess the utility, just say Here we use...

Line 21: has been claimed or has been suspected, not argued

Line 34-36 and line 310-322: "Third, within 20 x 20 km grid squares, the likelihood of satellite fixes being from the terminal week increased 10-fold from squares with no grouse moors, to those with 50 % cover of grouse moors." This is not clear to me. How did the distribution of fixes change? What was the proportion of fixes prior to the terminal week? Is the distribution of fixes in squares with no moors much lower whether or not in the terminal week?

This is an important point to clarify because I presume that most of the habitats for harriers are in cells with grouse moors so the change is what is important. Reword and revise for clarity.

Line 41: delete last sentence. This is not a techniques paper.

Line 108: data is plural thus say "data were received only. . ."

Line 135-136: whose and who are to be used for human subjects

Line 179: move sentence to end of paragraph

Line 199: delete took a conservative approach

Line 296: Do not begin sentence with a number—recast sentence.

Line 341: is shown also

Line 433: that were shot in the legs

Reviewer #1 (Remarks to the Author):

The paper examines the unaccountable disappearance of satellite-tracked harriers, and finds that most of the disappearances occur on grouse moors, where killing of harriers has long been suspected, accounting for the rarity of the species. This is an important paper, of interest to many people concerned with bird conservation and with the illegal killing of protected species. In my view the data are well analysed, with appropriate statistical treatment, and the conclusions are well justified.

My comments are mostly minor:

Line 145 reports re-sightings of birds with failed transmitters. Does this imply that birds were marked in other ways too? Explain.

All birds were ringed and a few were patagial tagged we have added text to say this (Methods – Satellite transmitters section):

“All nestlings were fitted with individual identity BTO (British Trust for Ornithology) metal rings and some tracked individuals were also fitted with patagial tags (n= 5).”

The two re-sighting of birds with failed transmitters were 1) from hide watches and camera traps at a nest (at Langholm) and 2) from digital photos showing a harrier with tag with a distinctive twisted aerial. In one case, the metal BTO ring was used to identify the individual bird using digital photography and in the other the bird was recognisable from a patagial tag. We have clarified this in the results section:

“Two birds were re-sighted after their tags had failed due to a malfunction (the individuals were recognisable by photography of the metal identity ring or patagial tag)”.

Line 195. Can you say, how long after a burn, strip fires are apparent from an aerial view? Readers might think they can be discerned for only a year or two after a burn, when in reality I guess it could be more than a decade. This affects the validity of your method of defining locations of grouse moors.

Thank you for pointing this out. Burnt areas can be visible for up to 25 years, and in addition to this, they are usually small and done on rotation. So any intensively managed grouse moor will always have mosaic of freshly burned through to old burned and unburned heather. Thus, it is always readily visible in some form by aerial photography. We have added some extra information to the text, which now reads:

“Burning of heather in small strips (up to 2 ha) in rotations of 10 to 25 years is a ubiquitous and widespread moorland management practice to support red grouse shooting, detected over at least 8551 km² of mainland UK^{34,57,58}. Post-burning regrowth is distinguishable from the surrounding unburned vegetation and the resultant patterns of burnt areas are easily detectable from remotely sensed images for up to ca. 25 years^{8,58,59}.”

Line 292. Can you say if any birds lasted a long time (say throughout the first winter) in a grouse moor area?

Most of the deaths or disappearances occurred in the first 20 weeks. During this time, the only individuals to survive were not spending much time on

grouse moors, thus no individuals lasted a long time on grouse moors. We have added the following sentence to summarize this:
“Only 20 individuals lasted through the first 20 weeks. During this period, the mean percentage of fixes on grouse moors per week for harriers that survived was (\pm SE) $15\% \pm 0.03$, which was half of the mean percentage for those which died or disappeared $30\% \pm 0.04$.

Line 344. Do you want to comment on the fact that some squares without grouse moors also showed relatively high terminal records? Is there anything you can say about these squares (eg wind turbines)?

We have added the following statement on this:

“A small number ($n=6$) of grid squares outside of areas managed for grouse moors also had a high proportion of terminal fixes. In most cases these squares directly bordered squares with managed grouse moors ($n=5$) and it is likely harriers were moving between squares. Only one square had a high proportion of terminal fixes but did not directly border a grouse moor and this can be attributed to the movements of just one individual.”

Line 392. 'dispersed around the country' seems a fairly loose statement. Looking at the map, there seems to be a southerly bias, as no birds appeared above the Edinburgh-Glasgow line.

This comment appears to be the result of a misunderstanding, driven by Figure 3. The area represented by this figure is not the only areas where birds went. There was movement northwards and one even went as far south as France! However, after constructing various maps of various coverage and areas, we choose this as the most sensible because it illustrates the area where the vast majority of the fixes from the different birds were located, thus movements in this area are best understood. We have now added some text into the legend of Figure 3 to reduce future misunderstanding about this aspect;

“Data includes only hen harriers that were known to have been illegally killed and those which suddenly disappeared when their tag stopped with no indication of a prior malfunction ($n=42$) **and the area (grid squares) encompassed 98% of the tracking data from these individuals and 91% of data from all harriers**”.

This area was selected for the illustrative figure, but all data in the UK were used for the analysis shown in Figure 2, we hope this is now clearer.

For other more minor comments, and suggestions on wording, see the copy of the ms attached.

Thank you, minor comments have also been addressed in the manuscript

Reviewer #2 (Remarks to the Author):

This is not a techniques paper but a paper identifying illegal killing of hen harriers. Certainly the study design does not allow a comparative analysis of the Argos collars or a systematic evaluation of radio performance. And such a techniques paper would not warrant publication in Nature Comms, but this is simply a matter of the authors having lost their way and is easily fixed.

Regardless, the results are important and we need such an analysis documenting the suspected killing of harriers by game keepers.

Thank you, we have made appropriate changes to the manuscript as outlined below and have also edited the manuscript title to more accurately reflect our work and main finding.

Line 18: Here we assess the utility, just say Here we use...

Edited

Line 21: has been claimed or has been suspected, not argued

Edited to claimed

Line 34-36 and line 310-322: "Third, within 20 x 20 km grid squares, the likelihood of satellite fixes being from the terminal week increased 10-fold from squares with no grouse moors, to those with 50 % cover of grouse moors." This is not clear to me. How did the distribution of fixes change? What was the proportion of fixes prior to the terminal week? Is the distribution of fixes in squares with no moors much lower whether or not in the terminal week?

This is an important point to clarify because I presume that most of the habitats for harriers are in cells with grouse moors so the change is what is important. Reword and revise for clarity.

We have edited the abstract extensively to reduce technical language. We hope that the briefer description of this finding is easier to understand: "At the landscape scale, satellite fixes from the last week of life were distributed disproportionately on grouse moors in comparison to the overall use of such areas."

To clarify the change in the distribution of fixes we have added a frequency plot (Supplementary Figure 2), which we now refer to in the main text "Fixes from the terminal week were distributed disproportionately on grouse moors compared to their overall use (Supplementary Figure 2)". We hope these measures have cleared any prior confusion.

Line 41: delete last sentence. This is not a techniques paper.

Agreed and deleted.

Line 108: data is plural thus say "data were received only. . ."

Edited

Line 135-136: whose and who are to be used for human subjects

Thank you, edited

Line 179: move sentence to end of paragraph

Moved

Line 199: delete took a conservative approach

Deleted

Line 296: Do not begin sentence with a number—recast sentence.

Thank you we have edited this sentence

Line 341: is shown also

Edited

Line 433: that were shot in the legs

Edited

REVIEWERS' COMMENTS:

Reviewer #1 (Remarks to the Author):

I have already commented on this paper, and the two comments below are intended as comments on your changes to the ms.

Original line 195. In describing patchy rotational muirburn, do you need both words 'ubiquitous and widespread' or is one enough?
If it is ubiquitous, it must be widespread.

Caption to Fig 3. Data are plural, so 'include' rather than 'includes'.

Reviewer #2 (Remarks to the Author):

I am pleased with the careful attention to reviewer comments. Thank you.
Mark Boyce

Response to Reviewers:

Thank you for the second review of our manuscript "**Patterns of satellite tagged hen harrier disappearances suggest widespread illegal killing on British grouse moors**". In response to the brief reviewers comments we are happy to confirm that we have made the two changes to the manuscript as suggested by Reviewer 1. These are:

Reviewer #1 (Remarks to the Author):

I have already commented on this paper, and the two comments below are intended as comments on your changes to the ms.

Original line 195. In describing patchy rotational muirburn, do you need both words 'ubiquitous and widespread' or is one enough?

If it is ubiquitous, it must be widespread.

- We have removed "and widespread".

Caption to Fig 3. Data are plural, so 'include' rather than 'includes'.

We are grateful this was noticed and have edited to 'include'.

Reviewer #2 (Remarks to the Author):

I am pleased with the careful attention to reviewer comments. Thank you.

Mark Boyce